# Diagnostic Errors in the Acutely Dizzy Patient—Lessons Learned

**DOI:** 10.3390/brainsci15010055

**Published:** 2025-01-09

**Authors:** Alexander A. Tarnutzer, Nehzat Koohi, Sun-Uk Lee, Diego Kaski

**Affiliations:** 1Neurology, Cantonal Hospital of Baden, 5404 Baden, Switzerland; 2Faculty of Medicine, University of Zurich, 8006 Zurich, Switzerland; 3Department of Neurology, National Hospital for Neurology and Neurosurgery, London WC1N 3BG, UK; n.koohi@ucl.ac.uk (N.K.); d.kaski@ucl.ac.uk (D.K.); 4The Ear Institute, University College London, London WC1X 8EE, UK; 5SENSE Research Unit, Department of Clinical and Movement Neurosciences, Queen Square Institute of Neurology, University College London, London WC1N 3BG, UK; 6Neurotology and Neuro-Ophthalmology Laboratory, Korea University Medical Center, Seoul 02841, Republic of Korea; sulee716@gmail.com; 7Department of Neurology, Korea University Medical Center, Seoul 02841, Republic of Korea

**Keywords:** acute vestibular syndrome, dizziness, headache, vertigo, diagnostic error, migraine, MRI, BPPV

## Abstract

Acute vertigo or dizziness is a frequent presentation to the emergency department (ED), making up between 2.1% and 4.4% of all consultations. Given the nature of the ED where the priority is triage, diagnostic delays and misdiagnoses are common, with as many as a third of vertebrobasilar strokes presenting with acute vertigo or dizziness being missed. Here, we review diagnostic errors identified in the evaluation and treatment of the acutely dizzy patient and discuss strategies to overcome them. Lessons learned include focusing on structured history taking, asking about timing and triggers to inform a targeted examination, assessing subtle ocular motor findings (e.g., by use of HINTS(+)), and avoiding overreliance on brain imaging (including early magnetic resonance imaging including diffusion-weighted sequences [DWI-MRI]). Importantly, up to 20% of DWI-MRI may be false negatives if obtained within the first 24–48 h after symptom onset. Likewise, overreliance on focal neurologic findings to confirm a stroke diagnosis should be avoided because isolated dizziness, vertigo, or even unsteadiness may be the only symptoms in some patients with vertebrobasilar stroke. Furthermore, in patients with triggered episodic vestibular symptoms provocation maneuvers should be preferred over HINTS(+), and a potential diagnosis of stroke should not be immediately dismissed in younger patients presenting with a headache (where migraine may be more common), but the possibility of a vertebral artery dissection should be further evaluated. Importantly, moderate training of non-experts allows for significant improvement in diagnostic accuracy in the acutely dizzy patient and thus should be prioritized.

## 1. Background

Acute vertigo and dizziness are amongst the most frequent presenting symptoms to the emergency department (ED), constituting between 2.1% and 4.4% of all admissions [1,2,3]. This translates to about 4.4 million consultations per year in the US (and probably 50 to 100 million worldwide [4]), resulting in estimated annual health care costs of over USD 10 billion in the US [5]. No single disorder accounts for more than 5% to 10% of all dizzy cases, increasing the risk for both inappropriate diagnostic testing and diagnostic errors [1]. About 15% of all patients presenting to the ED with acute vertigo, dizziness, or gait unsteadiness eventually are diagnosed with a sinister underlying cause [1]. The broad differential diagnosis spanning over almost all specialties, a lack of clinical experience in diagnosing dizzy patients, and the often transient nature of the patient’s complaints underscore the significant challenges faced by the busy ED physician. Amongst all acutely dizzy patients, about 3–5% will be eventually diagnosed with an ischemic stroke [4], with diagnostic criteria for vascular vertigo being published by the Bárány Society [6]. This fraction rises to about 25% for the subset of patients presenting with an acute vestibular syndrome (AVS) [7], i.e., acute and persistent vertigo or dizziness accompanied by nausea/vomiting, motion intolerance, and gait unsteadiness, and often also nystagmus [8]. In a systematic review on diagnostic errors in the ED it has been reported that 5.7% (95% confidence interval [CI] = 4.4–7.1%) of all ED visits had at least one diagnostic error [9]. Focusing on cerebrovascular events, it has been estimated that about 9% of all cerebrovascular events are missed at initial ED presentation [10]. The risk of misdiagnosis was much greater when presenting neurologic complaints were mild, non-specific, or transient. For non-specific symptoms (dizziness vs. motor findings) the false negative rate rose to 39.4% vs. 4.4% (odds ratio [OR] = 14.22 [CI = 9.76–20.74]), whereas for transient symptoms (transient ischemic attack [TIA] vs. ischemic stroke) the false positive rate was 59.7% vs. 11.7% (OR = 11.21 [CI = 6.66–18.89]) [10]. This is reflected also in a population-based study, demonstrating that ED misdiagnosis appears common for vertebrobasilar strokes presenting with acute dizziness, occurring in approximately 35% of cases [11]. In a German study, 44% of all acutely dizzy patients seen by ED consult neurologists were initially misdiagnosed [12]. Specifically, 6% of all benign ED diagnoses were corrected to sinister diagnoses, and 23% of all sinister ED diagnoses were revised to benign. Training background was found to have a significant impact on misdiagnosis rate, with stroke hospitalization after misdiagnosis of “benign dizziness” being lower in specialty care than in general practice. Missed stroke-related harms in general care were roughly twice those in specialty care in one study [13].

These numbers emphasize the need to improve diagnostic accuracy in acutely dizzy patients to minimize diagnostic errors and mitigate misdiagnosis-related harms. Different approaches have been proposed to address these shortcomings, including expanding access to neurologists in the ED [14], prioritizing specialized telemedicine consults, and expanding MR-imaging [10]. These strategies show promise; however, none of them are currently cost-effective and accessible in most EDs. Over the past 15 years, significant progress has been made in bedside diagnostic approaches for acutely dizzy patients. Introducing targeted bedside examination techniques focusing on ocular motor and vestibular signs, such as the HINTS in 2009 (Head Impulse, Nystagmus, Test of Skew [15]) and its expansion HINTS+ [16] in 2013, STANDING in 2015 [17] and graded truncal instability (GTI) ratings in 2006 (see [18] for review) have demonstrated high diagnostic accuracy [19] (see Appendix A for detailed description of these algorithms). Notably, even a few hours of dedicated formal training for ED physicians in examining acutely dizzy patients resulted in a very high stroke detection rate amongst AVS patients [20,21]. Recently, in the third edition of guidelines for reasonable and appropriate care in the emergency department (GRACE-3) consensus recommendations on how to address acute dizziness and vertigo in the emergency department have been published [22], providing a comprehensive review of the topic.

Despite these advancements, challenges persist in the diagnostic approach to acutely dizzy patients. In this critical review, we discuss common diagnostic misconceptions and errors in acutely dizzy patients and focus on the lessons learned. Diagnostic errors identified include overreliance on symptom quality, brain imaging, and obvious focal-neurologic findings, but also selecting the wrong examination techniques and the erroneous interpretation of those bedside findings retrieved. Finally, we outline ongoing challenges and discuss potential strategies to overcome these limitations.

## 2. Diagnostic Errors in Patients with Acute Dizziness

### 2.1. Diagnostic Error 1—Overreliance on Symptom Quality in the Acutely Dizzy Patient

Classic teaching [23] divides dizziness into four types based on symptom quality: vertigo (false sense of spinning or motion), presyncope, unsteadiness, and nonspecific or other types of dizziness. This reflects current practice in the United States [24], and in European countries, such as Switzerland, where 92% of primary care physicians consider it important to ask about the type of dizziness [25]. The underlying assumption is that the quality of dizziness symptoms predicts the principal cause: vertigo suggests a vestibular origin, presyncope a cardiovascular cause, disequilibrium a neurologic issue, and nonspecific dizziness a psychiatric or metabolic condition [26]. Such an approach influences subsequent diagnostic steps [27] as described in numerous textbooks and review articles [28]. However, a critical review of the literature suggested that this approach is not an evidence-based practice [28]. Two observations support this conclusion: first, the type of dizziness in acutely affected patients seems to be an imprecise metric, as more than half of the patients in a study of acutely dizzy individuals presenting to the ED were unable to reliably identify the symptom type that most accurately reflected their experience [2]. Second, and more importantly, the type of dizziness does not appear to reliably predict the underlying cause. While patients with acute unsteadiness were at a slightly increased risk of stroke, the presence of vertigo versus other types of dizziness was found to predict stroke with equal likelihood [11]. In a systematic review, dizziness caused by cardiovascular disease was often described as vertigo, contrasting with the concept proposed by Drachman and colleagues, which links presyncope to cardiovascular causes [29]. Third, healthcare professionals define terms such as vertigo, dizziness and unsteadiness inconsistently, further complicating this issue [2].

This limitation can be addressed by adhering to consensus definitions of vestibular symptoms and by shifting the focus onto the history of the acutely dizzy patient to features proven to aid in narrowing down the differential diagnosis. Established expert international consensus definitions for vestibular [30] and related symptoms [31] should therefore be considered. Structured history-taking is essential for both triaging, selecting appropriate bedside testing, and determining the need for additional diagnostic tests. This is especially important for patients presenting with vague symptoms such as vertigo, dizziness, or gait unsteadiness. The recently proposed TiTrATE approach (acronym for “timing, triggers, and targeted examinations”) by Edlow and Newman-Toker addresses these limitations [4] and offers a framework in which to classify every dizzy patient into one of three vestibular syndromes: acute vestibular syndrome (AVS), episodic vestibular syndrome (EVS), and chronic vestibular syndrome (CVS), as endorsed by an international committee of specialists [30]. By focusing on two key questions, namely, the timing of the patient’s vestibular symptoms and the presence or absence of triggers, the approach allows classification into one of six categories (see Table 1), and thus narrows down the differential diagnosis. Timing refers to the onset, duration, and evolution of dizziness, while triggers pertain to actions, movements, or situations that provoke symptoms in patients with intermittent dizziness [4]. For patients with suspected benign paroxysmal positional vertigo (BPPV), this approach has proven highly effective, as short-duration, triggered dizzy spells are the most reliable predictors of BPPV, despite substantial variability in the reported type of dizziness [32]. Similarly, a Swiss study reporting on the differential diagnosis of dizziness in ED patients found that more than one-third of unclear cases could be clarified by either assigning a vestibular syndrome diagnosis or excluding it altogether [33]. This clarification was achieved through an extended history obtained during follow-up, which focused specifically on the timing of symptom onset and the absence or presence of triggers. For acute and persistent vertigo or dizziness, a lack of triggers puts the emphasis on differentiating between vertebrobasilar stroke and acute unilateral vestibulopathy (AUVP) (see Table 2). History-taking should also include asking about vascular risk factors (arterial hypertension, dyslipidemia, diabetes, smoking, positive family history for vascular diseases), current medication, pre-existing medical conditions such as neoplasms or multiple sclerosis, recent head- or neck-trauma, and accompanying symptoms. These aspects may put more emphasis on one specific differential diagnosis, such as vertebral artery dissection in the case of a recent neck trauma or stroke if several vascular risk factors are present.

### 2.2. Diagnostic Error 2—Overreliance on Focal Neurologic Findings, i.e., Interpreting Isolated AVS as Peripheral in Origin

In acutely dizzy patients presenting with obvious focal neurologic signs and symptoms, associating their complaints with a central cause (mostly ischemic stroke) is usually straightforward. Vice versa, patients with isolated vertigo or dizziness are often presumed to have a peripheral cause and are therefore discharged without specific treatment or further diagnostic workup. However, a recent meta-analysis of high-quality studies involving relatively unselective AVS patient cohorts found that 66% of patients presenting with central AVS did not show any obvious focal neurologic signs [19]. The following signs were considered “obvious” in this meta-analysis: facial palsy, hemisensory loss, crossed sensory loss, dysphagia, dysarthria, limb ataxia, mental status abnormality (e.g., lethargy), hemiparesis, ocular motor paralysis, Horner syndrome, or visual field loss. Of note, acute unilateral hearing loss was conspicuously absent from this list and has long since been ‘lost’ to otolaryngology whereas, in fact, it may be of diagnostic use for vascular vertigo syndromes (see Section 2.5). Indeed, the absence of obvious focal neurologic signs does not exclude a central cause of the patient’s complaints. With the risk of the misdiagnosis of cerebrovascular events reportedly being much higher in patients with non-specific symptoms such as dizziness compared to motor symptoms (OR = 14.22 [CI = 9.76–20.74]) [10], this concept has often not been implemented in the diagnostic approach to the acutely dizzy patient by the ED physician. Isolated vertigo has been reported in 21% of all vertebrobasilar TIAs [44] and is the most common vertebrobasilar warning symptom before stroke [45]. However, a population-based registry found that vertebrobasilar TIA was not recognized at first medical contact in about 90% of cases [45], emphasizing the importance of linking both persistent and transient isolated vertigo/dizziness to potential central (ischemic) causes. The association between isolated vestibular symptoms and cerebrovascular events is also emphasized by the observation that patients hospitalized for isolated vertigo had a threefold higher risk of stroke over four years compared to those hospitalized for appendectomy (see [46] for an in-depth-discussion). Thus, patients with transient or persistent isolated dizziness should be assessed thoroughly for potential central causes as well as taking into account vascular risk factors and exploring ‘young stroke’ workup (including increased thrombotic risk as e.g., seen in antiphospholipid syndrome).

### 2.3. Diagnostic Error 3—Use of HINTS(+) in the Wrong Patients (Positional Vertigo, No Nystagmus)

If patients are selected appropriately, use of HINTS allows the identification of ischemic stroke with high diagnostic accuracy (sensitivity = 95.3%, specificity = 92.6%, see [19] for review). Importantly, the HINTS and HINTS+ have been validated for patients with a well-defined clinical presentation—i.e., an AVS (see Appendix A for details). This refers to patients with acute spontaneous and ongoing vertigo or dizziness that is accompanied by either spontaneous nystagmus (SN) or gaze-evoked nystagmus (GEN) [15]. In the original publication on HINTS, further patient selection, however, was made, requiring the presence of at least one vascular risk factor, and not all patients included presented with nystagmus [15]. With increasing popularity, HINTS(+) are now applied by frontline clinicians and specialists in a much more liberal way and outside of the patient populations for which HINTS(+) have been thoroughly validated. This includes applying HINTS(+) to patients with acute and ongoing vertigo/dizziness not accompanied by any nystagmus (either SN or GEN) and to patients with episodic (spontaneous or triggered) vertigo or dizziness outside of the episode. Consecutively, a drop in diagnostic accuracy of HINTS(+) in such patient populations is not surprising, as, e.g., seen by Kerber and colleagues when including acutely dizzy patients without nystagmus [36]. This is mainly due to the rule of considering an intact angular vestibulo-ocular reflex as a central sign in the AVS patient. Thus, when applying HINTS(+) to other patient populations with non-vestibular dizziness, they will be erroneously interpreted as central (i.e., head-impulse test normal, no gaze-evoked nystagmus, and no skew deviation) and specificity of HINTS(+) drops substantially. Along these lines, HINTS(+) is sometimes erroneously considered ‘normal’ or ‘abnormal’. However, when applied to the appropriate patient (with nystagmus), it should only be classified as either ‘peripheral’ or ‘central’.

Central AVS due to stroke is usually linked to the vertebrobasilar territory. However, in a prospective observational study, an incidence of acute vestibular symptoms in supratentorial stroke patients of 3.7% (48/1301, including 13 patients with isolated vertigo) has been reported [47]. These patients rarely presented with nystagmus (only 5/48), thus HINTS(+) could not be applied. However, impairments in pursuit eye movements were frequent in this subset (41%, 18/44). Thus, assessing ocular motor properties may be a potential approach to identify a central cause in such patients [48] but will require additional training for non-experts.

These observations emphasize the need for training of frontline clinicians and specialists in selecting the right patients for HINTS(+) testing, applying the test correctly, and interpretating the findings appropriately. It has been demonstrated that expert neuro-otologists achieve higher specificity with HINTS testing compared to non-specialists (0.98 [CI = 0.95–1.00] vs. 0.89 [CI = 0.83–0.95]) [19]. However, the sensitivity between these two groups is comparable, confirming that the risk of missing a stroke (i.e., the false negative rate) remains low among non-experts. For patients with acute prolonged vertigo or dizziness who do not present with any nystagmus, HINTS(+) testing should be combined with a graded assessment of truncal instability to identify those with an acute imbalance syndrome (AIS) and with positional testing and therefore to increase the specificity of HINTS(+) in this setting. Importantly, patients unable to sit or stand unassisted have a very high probability (99.1%) of an underlying central cause, although sensitivity for this sign is moderate (44.0%) [18]. Alternatively, the STANDING algorithm may be applied to these patients [17] (see Appendix A). This approach assigns patients without nystagmus but with persistent vertigo or dizziness directly to postural stability assessment, bypassing tests for subtle ocular motor findings. It is noteworthy that this algorithm classifies patients unable to stand or walk independently as having a central cause, aligning with a grade 3 truncal instability rating [49]. Furthermore, the STANDING algorithm does not include hearing testing and also does not provide any approach for how to distinguish peripheral from central positional nystagmus.

HINTS(+) has been considered a tool for specialized neuro-otologists due to the subtlety of its findings and clinical application, and it often been deemed unsuitable for use in the ED setting to rule out stroke with sufficient accuracy [50]. However, recent studies have demonstrated that ED physicians can successfully learn to apply HINTS with high diagnostic accuracy after just few hours of training. In a French study, ED physicians achieved high diagnostic accuracy following six hours of formal training, which included four hours of lectures and two hours of workshops [21]. Another study demonstrated that training ED physicians using a virtual-reality-enhanced mannequin significantly improved diagnostic performance, with these skills retained for at least six months after one hour of simulator training [51]. These findings demonstrate that with dedicated training lasting two to six hours, frontline clinicians and other non-specialists can apply HINTS(+) with high diagnostic accuracy.

### 2.4. Diagnostic Error 4—Assuming That Worsening Symptoms During Positional Testing Definitely Indicate BPPV

In acutely dizzy patients, worsening of symptoms during positional testing is often considered proof for BPPV, leading to the application of repositioning maneuvers. Improvement is typically assessed during follow-up consultations. However, these patients risk missing a stroke diagnosis if further evaluation for potential central (ischemic) causes is not pursued in cases of BPPV misdiagnosis. Importantly, the distinction must be made between triggering new symptoms (provocation) and worsening existing symptoms (exacerbation). In AVS, symptoms are exacerbated by positional changes, whereas in BPPV, symptoms are provoked. A diagnosis of BPPV is justified only if positional changes trigger symptoms and are accompanied by typical nystagmus (e.g., upbeat and torsional for posterior-canal BPPV) (for diagnostic criteria published by the Bárány Society see [52]). Worsening of pre-existing symptoms alone does not support a BPPV diagnosis. Furthermore, isolated paroxysmal positional vertigo and nystagmus may occur in central cases, e.g., due to small ischemic lesions around the fourth ventricle or the cerebellum [53]. It is explained by disinhibition of irregular afferent signals transmitted to the vestibular nucleus, generating a prominent post-rotatory response [54]. Nystagmus patterns atypical for BPPV (e.g., nystagmus evoked in multiple planes), accompanying central signs, and a lack of response to treatment should raise suspicion for a central cause and trigger further diagnostic workup.

### 2.5. Diagnostic Error 5—Assuming That Auditory Symptoms in Acutely Dizzy Patients Always Imply a Peripheral Cause

In acutely dizzy patients, hearing loss is often attributed to an inner ear problem, presenting as a combined acute audio-vestibular unilateral loss. It is essential to actively search for and exclude ear pain and abnormalities of the eardrum. Notably, new-onset unilateral hearing loss does not align with a diagnosis of acute unilateral vestibulopathy according to the Bárány Society diagnostic criteria [55]. While a combined audio-vestibular loss may occur with a painful ear, as seen in labyrinthitis or Ramsey–Hunt syndrome, new-onset, painless unilateral hearing loss should raise suspicion for a central cause. It has been demonstrated that in AVS patients with peripheral HINTS findings, a new-onset, unilateral hearing loss on the side of the abnormal head-impulse test strongly predicts a vascular cause, such as a labyrinthine stroke or lateral pontine stroke [16]. This is especially relevant for strokes in the territory of the anterior inferior cerebellar artery (AICA), where adding new-onset hearing loss as a fourth sign to HINTS (referred to as HINTS(+)) increases the sensitivity from 84.0% (CI = 65.3–93.6%) to 95.7% (CI = 79.0–99.2%) (see Figure 1 for a case description). Thus, in patients with AVS and peripheral HINTS findings, hearing loss should be actively assessed at the bedside by using of finger rub, whispered words, a tuning fork, or other low-tech hearing screening tools (see below for further discussion on the hearing screening tool). If hearing loss is present and not accompanied by ear pain or a red eardrum, this should raise suspicion of a stroke. Moving forwards, there may be scope to introduce hand-held screening audiometers or even smartphone Apps into the ED to address this diagnostic gap. Furthermore, the presence of a bilaterally positive head-impulse test in cases of acute audio-vestibular unilateral loss [56] or central-type head-shaking nystagmus (HSN, i.e., perverted HSN, HSN in the opposite direction of spontaneous nystagmus, HSN beating towards unilateral canal paresis, or abnormal head-impulse-testing) [57] may also indicate a stroke.

### 2.6. Diagnostic Error 6—Assuming a Negative CT Scan Excludes an Ischemic Stroke

Head CT is generally not helpful in evaluating acutely dizzy patients. Numerous studies have demonstrated the limited value of non-contrast CT imaging [58,59] and CT-angiography (CT-A) in patients with acute dizziness or vertigo. For example, 94.1% (CI = 89.4–96.7%) of dizziness visits with a CT did not receive a CNS diagnosis [60], and in only 1.3% of patients with isolated dizziness that received CT-A, did examinations result in a change in clinical management [61]. Cerebellar or other hemorrhages rarely present with isolated dizziness or vertigo; these cases almost always include other symptoms such as mental status change, dysarthria, severe headache, or hemiparesis [62]. As such, CT is not justified for ruling out intracerebral hemorrhage in the absence of general neurological examination abnormalities [19]. Likewise, brain imaging is unnecessary for triggered EVS (i.e., indicating probably BPPV), except where the nystagmus pattern is atypical for BPPV and treatment response is lacking [63].

In a recent meta-analysis on the diagnostic value of non-contrast CT-imaging in the AVS patient, a sensitivity of 28.5% (CI = 14.4–48.5%) for detecting a stroke was reported [38], whereas sensitivity of DWI-MRI in acutely dizzy patients for detecting a stroke was 79.8% (CI = 71.4–86.2%).

The indications for emergency CT head scans (including CT-A)—other than accessibility—for acute dizziness/vertigo are a high NIHSS score in AVS (which would imply the presence of other neurological features beyond vertigo) or clinically relevant deficits within the treatment window for intravenous thrombolysis (IVT, on-label treatment within 4.5 h after symptom onset, treatment possible up to 9 h after last-seen normal in selected cases [64]), or endovascular treatment (EVT, possible up to 24 h after last-seen normal [65]). CT-perfusion for infratentorial stroke may be helpful in identifying large cerebellar stroke, but the resolutions are usually too low to allow detection of hypoperfusion in the brainstem. Traditionally, it has been difficult to convince radiologists to progress straight to MR imaging without CT, but now there is sufficient evidence to back this clinical decision for acute vertigo.

### 2.7. Diagnostic Error 7—Assuming a Negative DWI-MRI Excludes an Ischemic Stroke

While brain DWI-MRI has significantly higher sensitivity than non-contrast brain CT imaging for detecting vertebrobasilar stroke, it has a false negative rate of up to 20.2% [CI = 13.8–28.6%] when performed within the first 24–48 h after symptom-onset in acutely dizzy patients presenting to the ED [38]. When presenting with AVS, false negative rates vary based on stroke size, reaching 53% for small brainstem strokes [34] and approximately 15% for pooled central AVS cases [19]. As a gold-standard, DWI-MRI should be obtained with a delay of 3–5 days after symptom-onset, as described in the case presented in Figure 2 [66]. Thus, in AVS patients with an initial bedside neuro-otologic assessment being indicative of a central cause (i.e., HINTS(+) are central and/or severe truncal instability is observed) and early DWI-MRI being negative, the patient should be transferred to the stroke unit for monitoring and further stroke-workup. A repeat DWI-MRI 3–5 days after symptom onset is recommended for confirmation. This approach is supported by the substantially higher sensitivity of bedside testing with algorithms such as HINTS(+) (97.2% [CI = 94.0–100.0%] [19]) and STANDING (93.4% [CI = 84.5–97.6%] [21]) compared to early DWI-MRI (85.1% [CI = 79.2–91.0%] [19]).

### 2.8. Diagnostic Error 8—Assuming That Acute Vertigo in Younger Patients Is Rarely a Stroke

About 10–15% of all strokes occur in people aged between 18 and 50 years, with a higher incidence in women than in men [67]. Stroke patients aged 18–44 years are 7 times more likely to be misdiagnosed compared to those aged 75 years or more [39]. Likewise, strokes presenting with acute vertigo or dizziness are missed more frequently in younger patients (aged < 50 years), women, and individuals from ethnic minorities [39,68]. Dizziness is also the most common presenting symptom of vertebral artery dissection (being observed in 58% of cases) [40]. This condition, which predominantly affects younger patients, often mimics migraine, and is therefore easily overlooked [41]. Thus, young age and the absence of vascular risk factors should not exclude a differential diagnosis of stroke in acutely dizzy patients and should trigger the TiTrATE approach including testing for subtle oculomotor findings [69]. Likewise, relying on the ABCD2 vascular risk score (including age, blood pressure, clinical features of unilateral weakness or speech change, duration of symptoms, diabetes) to identify a stroke in young, acutely dizzy patients poses its risks. While the use of the ABCD2 score has been promoted in acutely dizzy patients [70], it has been shown to have a low diagnostic accuracy in the acutely dizzy patient to predict a central cause (sensitivity = 61.1% (CI = 52–70%); specificity = 62.3% (CI = 51–72%) for an ABCD2-score of 4 or more) [16]. Thus, the value of the ABCD2 score in the setting of the acutely dizzy patient is limited.

### 2.9. Diagnostic Error 9—Assuming That Acute Vertigo with Headache Always Indicate Vestibular Migraine

Both migraine and acute vertigo/dizziness are frequent complaints in the general population. Specifically, about 18% of women and 6% of men suffer from migraines [71], and the estimated prevalence for vestibular migraine is up to 2.7% [72]. Diagnostic criteria for vestibular migraine have been published by the Bárány Society [73]. Likewise, surveys indicated a 1-year prevalence for vertigo or dizziness in the general population of up to 48.3% [74]. Thus, it may be tempting to attribute acute vertigo or dizziness with accompanying headaches to vestibular migraine, especially in young female patients (see diagnostic error 8). However, headache associated with ischemic stroke occurs frequently (7.4–34%), particularly in younger patients and those with vertebrobasilar strokes [42]. Vertebral artery dissection, a leading cause of stroke in the young [75], may also present with vertigo, headache, and neck pain. Contrastingly, migrainous headaches are typically recurrent, occurring more than five times, and often begin in the second or third decade of life.

In patients presenting with acute vertigo or dizziness and accompanying headaches, a thorough assessment of the timing, recurrence, and distribution of the ongoing headache is needed, in addition to reviewing the patient’s history for headaches. Note also that many patients with an acute episode of vestibular migraine will not experience headaches concurrently with the vertigo or dizziness. A unilateral, new-onset occipital headache or lateralized neck pain should trigger suspicion for possible vertebral artery dissection and result in CT/MRI brain imaging including angiography.

### 2.10. Diagnostic Error 10—Isolated Postural Instability Is Not a TIA/Stroke Symptom

Acute truncal ataxia may be the sole clinical finding in vertebrobasilar stroke/TIA and may make up to 15% of all patients presenting with acute vertigo to the ED [43]. Thus, the absence of nystagmus or subtle ocular motor findings does not exclude the possibility of a stroke in such patients (see case description in Figure 2) [76]. The terms acute imbalance syndrome (AIS) and acute truncal ataxia (ATA) have recently been coined to describe this condition [43]. Previously, a graded rating of gait and truncal instability (GTI) has been proposed, distinguishing gait imbalance while walking independently (grade 1) from inability to walk unassisted (grade 2) and from inability to stand/sit unassisted (grade 3) [49]. An acute imbalance syndrome with severe truncal instability (i.e., inability to stand/sit unassisted) is highly predictive for a central (ischemic) cause (specificity 99.1% [CI = 98.0–100.0%]) as recently summarized in a meta-analysis [18], but the sensitivity remains low (44% [CI = 34.3–53.7%]) for a stroke. Considering either grade 2 or grade 3 GTI as central signs, specificity (82.7% [CI = 71.6–93.8%]) and sensitivity (70.8% [CI = 59.3–82.3%]) are moderate. Thus, while such a graded GTI rating does not have a diagnostic accuracy as high as HINTS(+), it is a valuable addition in those dizzy/ataxic patients with no nystagmus. Clinicians are encouraged to formally assess standing balance in patients with acute vertigo, especially those hesitant to attempt standing).

## 3. Conclusions

The diagnostic approach to acutely dizzy patients has evolved significantly over the last 15 to 20 years. New bedside diagnostic algorithms have been implemented to overcome the limited sensitivity of the general neurologic exams in detecting strokes in these patients. This includes both the use of structured history-taking focusing on timing and triggers and a targeted examination [4] instead of the quality of dizziness for a first approximation of the differential diagnosis. With moderate training, use of dedicated bedside algorithms such as HINTS(+) or STANDING results in a diagnostic accuracy even surpassing that of early DWI-MRI [19,21]. Thus, in such cases, bedside neuro-otologic assessments indicative of a central cause (especially ischemic stroke) overrule negative early (i.e., within first 24–48 h) brain CT or DWI-MRI and support referral to a stroke unit. Importantly, central HINTS(+) may trigger acute stroke treatment in those cases with clinically relevant findings that present within the treatment window for intravenous thrombolysis and/or endovascular thrombectomy. It is noteworthy that the NIHSS (National Institutes of Health Stroke Scale) score may be low or even zero in vertebrobasilar strokes, but, nonetheless, clinical findings may have a substantial impact on daily living (such as severe postural instability and tendency to fall or oscillopsia due to gaze-evoked nystagmus or severe hearing loss) and thus may justify acute treatment [77]. Importantly, young age, female gender, and new-onset lateralized neck pain/headache should trigger suspicion of vertebral artery dissection. While the appropriate use of the techniques and approaches described here require training, it has been shown that moderately trained non-specialists, such as ED physicians, successfully apply the HINTS or STANDING algorithm. Thus, dedicated and structured teaching to frontline providers is strongly recommended. With resources for dedicated training being limited and with a high turnover of ED staff, complementary emerging trends, such as AI-assisted triaging tools, should be considered as well. Eventually, the combination of these approaches may result in a reduction in delayed or missed diagnoses and may positively affect the patient’s outcome.

## Figures and Tables

**Figure 1 brainsci-15-00055-f001:**
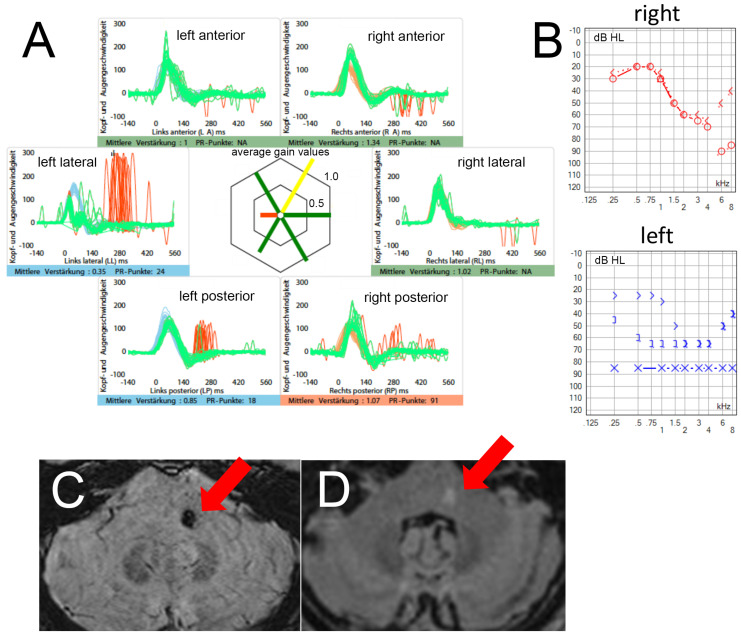
This patient presented with acute onset vertigo and nausea to the ED. On initial examination, he demonstrated a spontaneous horizontal nystagmus to the right. No focal neurologic signs were noted. HINTS were peripheral with a pathological bedside head-impulse test to the left (see panel (**A**) for video-head-impulse test), no evidence for gaze-evoked nystagmus, and absent skew deviation. However, the patient reported a new-onset hearing loss (HL) on the left side (confirmed on pure tone audiogram as demonstrated in panel (**B**)), resulting in central HINTS+. Thus, he was referred to the stroke unit for further diagnostic workup. On MRI of the brain, a focal hemorrhage (panel (**C**), susceptibility weighted imaging [SWI], red arrow) with FLAIR-positivity (panel (**D**), red arrow) at the border to the left cerebellar peduncle was found. A diagnosis of a cavernoma was made and a recent bleed was postulated to explain the patient’s current complaints. Conservative treatment was chosen and the patient eventually fully recovered. For video-head-impulse testing (in panel (**A**)) eye velocity traces (in green) and head velocity traces (in red for testing the right vestibular organ and in blue for assessing the left vestibular organ) are plotted against time. Note that eye velocity traces were inverted for better visualization and comparison with the head velocity traces and that gain was calculated as the ratio of the area under the de-saccaded eye-velocity curve to the area under the head-velocity curve, corresponding to a de-saccaded position gain. Summary plots in the center illustrate average individual vestibulo-ocular reflex (VOR)-gains ± 1SD for all six canals. Pure tone audiometry (panel (**B**)) symbols: red “o” = unmasked air-conduction testing of right ear; blue “x” = unmasked air-conduction testing of the left ear; red “<” = unmasked bone-conduction testing (on mastoid) of right ear; blue “>” = unmasked bone-conduction (on mastoid) testing of left ear; blue “]”= masked bone-conduction testing (on mastoid) of left ear.

**Figure 2 brainsci-15-00055-f002:**
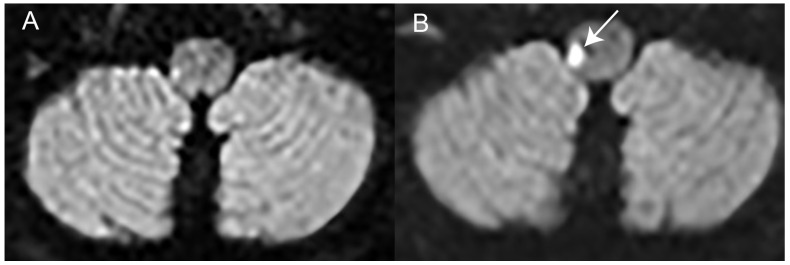
Early DWI-MRI negative stroke presenting as isolated severe truncal instability. This 93-year-old male patient presented with acute-onset gait imbalance and tendency to fall to the ED. He did not report any double vision, vertigo/dizziness, or dysphagia. On initial neurological assessment in the ED, he could sit unassisted; however, when standing, he showed a tendency to fall to the right side and had to be held. He did not demonstrate any pathological nystagmus patterns, cranial nerve abnormalities, palsies, or paresthesia. Thus, no HINTS+ were applied. On initial brain CT (including CT-angiography), no signs of acute stroke or vessel occlusions could be seen. With a suspicion of a potential stroke, the patient was referred to the stroke unit for further workup. About 36 h after symptom onset, an DWI-MRI of the brain demonstrated no acute ischemic lesions (panel (**A**)). With a high suspicion for ischemic stroke based on the clinical assessment demonstrating severe truncal instability (grade 3), DWI-MRI was repeated 72 h after symptom onset. Now an acute ischemic lesion in the right lateral medulla could be depicted (panel (**B**), white arrow), confirming a focal brainstem stroke. No cause for the patient’s stroke could be identified (TOAST 5). The patient was eventually referred to neurorehabilitation.

**Table 1 brainsci-15-00055-t001:** Vestibular syndromes * according to the TiTraTE approach [4].

Timing	Obligate Triggers Present ^†^	No Obligate Triggers ^†^
New, episodic	t-EVS (e.g., BPPV or orthostatic hypotension)	s-EVS (e.g., cardiac arrhythmia, vestibular migraine, Menière’s disease, vestibular paroxysmia, TIA)
New, continuous	t-AVS (e.g., post gentamicin, AED intoxication, traumatic unilateral vestibulopathy)	s-AVS (e.g., vertebrobasilar stroke, acute unilateral vestibulopathy, Wernicke encephalopathy)
Chronic, persistent	t-CVS (e.g., uncompensated unilateral vestibular loss, present only with head movement)	s-CVS (e.g., chronic, persistent dizziness associated with cerebellar degeneration or PPPD)

Abbreviations: AED = antiepileptic drug; BPPV = benign paroxysmal positional vertigo; PPPD = persistent postural–perceptual dizziness; s-AVS = spontaneous acute vestibular syndrome; t-AVS = traumatic/toxic acute vestibular syndrome; s-CVS = spontaneous chronic vestibular syndrome; t-CVS = context specific chronic-vestibular syndrome; s-EVS = spontaneous episodic vestibular syndrome; t-EVS = triggered episodic vestibular syndrome; TIA = transient ischemic attack. * Note that the use of the word vestibular connotes vestibular symptoms (dizziness, vertigo, imbalance, or lightheadedness and so forth) rather than underlying vestibular causes (e.g., benign paroxysmal positional vertigo or acute unilateral vestibulopathy). ^†^ Trigger for nonspontaneous forms refer to obligate triggers (episodic), exposures (acute, continuous), and contexts (chronic) that sharply distinguish these forms from their spontaneous counterparts. Spontaneous causes, as defined here, sometimes have underlying predispositions or precipitants, but these are not only-and-always associations.

**Table 2 brainsci-15-00055-t002:** Diagnostic errors in acutely dizzy patients (modified after [34]).

Diagnostic Error	Wrong Assumption	Solution	Remarks
Overreliance on the symptom quality in the acutely dizzy patient	The type of dizziness predicts the underlying cause (vertigo is vestibular, presyncope is cardiovascular, disequilibrium is neurologic, and nonspecific dizziness is psychiatric or metabolic	Focus on timing and triggers rather than on the type of dizziness [4]	Patients are inconsistent in describing the type of dizziness [2] Healthcare professionals define terms such as vertigo, dizziness, and unsteadiness inconsistently [2]. Cardiac arrhythmia may present as true vertigo [35]
Overreliance on focal neurologic findings	Absence of focal neurologic findings in the acutely dizzy patient indicates a peripheral cause	Focus on subtle ocular motor signs (HINTS+ (head-impulse test, nystagmus, test-of-skew, hearing [16]) or STANDING [17]) and truncal instability [18]	Focal neurologic findings are absent in about 2/3 of cAVS [19] AICA strokes are at highest risk for misdiagnosis—adding the 4th sign (new-onset unilateral hearing loss) to HINTS substantially increases sensitivity from 84.0% to 95.7% [19]
Use of HINTS in the wrong patients	HINTS are validated for all dizzy patients	Add a graded gait and truncal instability rating [18] to HINTS(+) in those patients with acute ongoing vertigo but no (spontaneous/gaze-evoked) nystagmus Be aware of low diagnostic accuracy of HINTS(+) in patients with transient AVS or triggered EVS (e.g., BPPV).	HINTS(+) have been validated only for AVS patients with nystagmus [16], and were often restricted to populations with at least one vascular risk factor [15,16] If used outside of this setting, diagnostic accuracy drops substantially [36]
Overreliance on positional testing in the AVS patient	Worsening of symptoms in positional testing confirms BPPV	Distinguish triggers (i.e., movement provokes symptoms) and exacerbating factors (i.e., movement makes existing symptoms worse).	Acute vertigo or dizziness is usually exacerbated by head movements. This is true both for peripheral and central causes [7] In BPPV, positional changes result in onset of transient vertigo or dizziness CPPV may mimic BPPV, features such as atypical nystagmus characteristics (beating direction), focal neurologic findings, and lacking response to liberation maneuvers may help identify central causes.
Using the presence of auditory symptoms to exclude a central cause	Acute hearing loss in the dizzy patient is linked to an inner-ear disorder	Be aware of vascular causes in acute unilateral auditory symptoms in AVS.	A significant fraction of AICA strokes present with new-onset unilateral hearing loss. Identified as central if HINTS+ are used [16] Labyrinthine stroke may result in hearing loss and tinnitus also [37] Lack of ear pain and a red ear drum make a diagnosis of labyrinthitis unlikely
Overreliance on brain CT imaging	A negative CT scan excludes an ischemic stroke	Be aware of the limitations of CT brain imaging	Sensitivity of non-contrast CT for detecting acute ischemic stroke is only 28.5% (14.4–48.5%) [38] Sensitivity of CT angiography for detecting acute stroke is even lower (14.3% [1.8–42.8%])
Overreliance on brain DWI-MRI imaging	A negative DWI-MRI excludes an ischemic stroke	Be aware of the limitations of DWI-MRI brain imaging	Early DWI-MRI (obtained in first 24–48 h) may be negative in 20.2% of acutely dizzy patients [38] and in 14.7% of AVS patients [19] For small lacunar strokes false-negative rate may be as high as 53% [34]
Overreliance on age for excluding stroke	Acute vertigo in younger patients is rarely a stroke	Do not overfocus on age and vascular risk factors. Consider vertebral artery dissection in young patients	Stroke patients aged 18–44 years are 7-fold more likely to be misdiagnosed than patients aged 75 years or more [39] Dizziness is the most common presenting symptom of VAD (58%) [40], which affects younger patients, mimics migraine, and is easily misdiagnosed [41]
Overreliance on headaches to confirm for migraine	Headache accompanying acute vertigo suggests vestibular migraine	Obtain a detailed description of headache characteristics and accompanying symptoms	Sudden, severe or sustained head or neck pain may indicate a (ruptured) aneurysm, dissection or other vascular pathology. Headache attributed to ischemic stroke is frequent (7.4–34%) and has a greater chance of occurring in younger patients and in vertebrobasilar strokes [42]
Discarding isolated acute truncal ataxia as central sign	Isolated postural instability is not a TIA/stroke symptom	Obtain a graded rating of truncal instability and consider stroke, especially if instability is severe [18]	Absence of nystagmus and/or focal neurologic signs does not exclude stroke as acute truncal ataxia may present in isolation [43] Inability to stand or sit unassisted is highly suggestive of a central (ischemic) cause of the patient’s isolated truncal ataxia [18].

Abbreviations: AICA = anterior inferior cerebellar artery; AVS = acute vestibular syndrome; BPPV = benign paroxysmal positional vertigo; cAVS = central AVS; CPPV = central paroxysmal positional vertigo; CT = computed tomography; DWI-MRI = magnetic resonance imaging with diffusion-weighted imaging; EVS = episodic vestibular syndrome; HINTS = Head-Impulse testing, Nystagmus, Test of Skew; HINTS+ = HINTS with additional testing for new-onset unilateral hearing loss; STANDING = SponTAneous Nystagmus, Direction, head Impulse test, standiNG); TIA = transient ischemic attack; VAD = vertebral artery dissection.

## Data Availability

No new data were created or analyzed in this study. Data-sharing is not applicable to this article.

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
