# Peer review of "Diagnostic Errors in the Acutely Dizzy Patient—Lessons Learned"

_brainsci, 2025, doi:10.3390/brainsci15010055_

Round 1

Reviewer 1 Report

Comments and Suggestions for Authors

Comments to authors

This article describes the diagnostic errors in the acutely dizzy patient and provides an excellent review of the diagnosis of dizziness, acute and chronic vestibular syndrome. “Lessons learned” are instructive.

The diagnostic approach to acute dizzy patients is very important. I completetely agree with the authors that the most important thing is to avoid misdiagnosing with cerebrovascular (ischemic) disease.

I do not think there’s any problem with the specifics. However, diagnostic criteria have been established for some acute vertigo/dizziness ( for example, vestibular migraine; J Vestibu Res 2012:167-172, BPPV; J Neurol 2021: 1995-2000)

While the contents of this review suggest that we should not be bound by the diagnostic criteria, I believe it is necessary to know the diagnostic criteria especially for young doctors just starting the training.

It may be a good idea to add papers on diagnostic criteria to the references.

Author Response

This article describes the diagnostic errors in the acutely dizzy patient and provides an excellent review of the diagnosis of dizziness, acute and chronic vestibular syndrome. “Lessons learned” are instructive.

The diagnostic approach to acute dizzy patients is very important. I completely agree with the authors that the most important thing is to avoid misdiagnosing with cerebrovascular (ischemic) disease.

I do not think there’s any problem with the specifics. However, diagnostic criteria have been established for some acute vertigo/dizziness ( for example, vestibular migraine; J Vestibu Res 2012:167-172, BPPV; J Neurol 2021: 1995-2000). While the contents of this review suggest that we should not be bound by the diagnostic criteria, I believe it is necessary to know the diagnostic criteria especially for young doctors just starting the training.

It may be a good idea to add papers on diagnostic criteria to the references

Reply by the authors: As proposed by the author we have added citations referring to the diagnostic criteria published (vestibular migraine, vascular vertigo, BPPV).

Reviewer 2 Report

Comments and Suggestions for Authors

Thank you for the opportunity to review the manuscript “Diagnostic errors in the acutely dizzy patient – lessons learned”, aimed to review diagnostic errors identified in the evaluation and treatment of the acutely dizzy patient and discuss strategies to overcome them”.

The topic is interesting for a broad readership. However, some aspects could be improved.

Supplementary material to include the algorithms described is required.

The reader will appreciate comments on the general clinical history (i.e. comorbidity, medication, risk factors) that would be particularly relevant for stroke.

No mention is given on the  Guidelines for reasonable and appropriate care in the emergency department 3 (GRACE-3): Acute dizziness and vertigo in the emergency department    from the Society for Academic Emergency Medicine.

Apart from the discussion provided, the reader would appreciate a Table describing the sensitivity and sensibility of the variety of clinical and imaging evaluations, according to the main peripheral and central classification of disorders.

Author Response

The topic is interesting for a broad readership. However, some aspects could be improved.

Reply by the authors: We thank the reviewer for his/her overall positive feedback and the helpful input for further improving the manuscript.

Supplementary material to include the algorithms described is required.

Reply by the authors: We have discussed both the HINTS+ and STANDING algorithm in our review. These two algorithms are now included in the supplementary material. Furthermore, we also provide a table with the detailed GTI rating.

The reader will appreciate comments on the general clinical history (i.e. comorbidity, medication, risk factors) that would be particularly relevant for stroke.

Reply by the authors: We have added general clinical history findings to the first diagnostic error:

“History taking should also include asking about vascular risk factors (arterial hypertension, dyslipidemia, diabetes, smoking, positive family history for vascular diseases), current medication, pre-existing medical conditions such as neoplasms or multiple sclerosis, recent head- or neck-trauma and accompanying symptoms. These aspects may put more emphasis on one specific differential diagnosis such as vertebral artery dissection in the case of a recent neck trauma or stroke if several vascular risk factors are present.“

No mention is given on the Guidelines for reasonable and appropriate care in the emergency department 3 (GRACE-3): Acute dizziness and vertigo in the emergency department    from the Society for Academic Emergency Medicine.

Reply by the authors: We thank the reviewer for pointing this out. We now refer to the recent GRACE-3 guideline in introduction:

“Recently, in the third edition of guidelines for reasonable and appropriate care in the emergency department (GRACE-3) consensus recommendations on how to address acute dizziness and vertigo in the emergency department have been published [1], providing a comprehensive review on the topic.»

Apart from the discussion provided, the reader would appreciate a Table describing the sensitivity and sensibility of the variety of clinical and imaging evaluations, according to the main peripheral and central classification of disorders.

Reply by the authors: We have added a figure from a recent publication from our group (Martinez et al. 2024, PMID 38990511) comparing the diagnostic accuracy of several bedside examination techniques (HINTS, HINTS+, clinical neuro-exam, graded truncal instability [GTI], GTI including central-type spontaneous nystagmus) with imaging (CT, MRI) to the supplementary material (Figure S2).

Round 2

Reviewer 2 Report

Comments and Suggestions for Authors

The manuscript has been improved. but no supplementary material was available in the document for review.